# Computer Vision-Based Inspection System for Worker Training in Build and Construction Industry

**M. Fikret Ercan * and Ricky Ben Wang**

Singapore Polytechnic, School of Electrical and Electronic Engineering, 500 Dover Road,
Singapore 139651, Singapore; rickywang04@gmail.com
* Correspondence: mfercan@sp.edu.sg

**Abstract:** Recently computer vision has been applied in various fields of engineering successfully ranging from manufacturing to autonomous cars. A key player in this development is the achievements of the latest object detection and classification architectures. In this study, we utilized computer vision and the latest object detection techniques for an automated assessment system. It is developed to reduce the person-hours involved in worker training assessment. In our local building and construction industry, workers are required to be certified for their technical skills in order to qualify working in this industry. For the qualification, they are required to go through a training and assessment process. During the assessment, trainees implement an assembly such as electrical wiring and wall-trunking by referring to technical drawings provided. Trainees' work quality and correctness are then examined by a team of experts manually and visually, which is a time-consuming process. The system described in this paper aims to automate the assessment process to reduce the significant person-hours required during the assessment. We employed computer vision techniques to measure the dimensions, orientation, and position of the wall assembly produced hence speeding up the assessment process. A number of key parts and components are analyzed and their discrepancies from the technical drawing are reported as the assessment result. The performance of the developed system depends on the accurate detection of the wall assembly objects and their corner points. Corner points are used as reference points for the measurements, considering the shape of objects, in this particular application. However, conventional corner detection algorithms are founded upon pixel-based operations and they return many redundant or false corner points. In this study, we employed a hybrid approach using deep learning and conventional corner detection algorithms. Deep learning is employed to detect the whereabouts of objects as well as their reference corner points in the image. We then perform a search within these locations for potential corner points returned from the conventional corner detector algorithm. This approach resulted in highly accurate detection of reference points for measurements and evaluation of the assembly.

**Keywords:** computer vision; deep learning; corner detection; quality construction

## 1. Introduction

Computer vision has long been applied for inspection and quality control tasks in various industries [1–4]. These applications are primarily concerned with detecting patterns, irregularities, or deriving information from the objects analyzed that can be broadly defined as recognition tasks. Among these, manufacturing is a typical application where geometric measurement tests for manufactured parts and artifacts are performed. Typically inspection involves automatic recognition of the object and then analyzing its geometric measurements. Using computer vision in manufacturing saves time, minimizes errors, and improves the automation process. It is important to note that in these applications, the environment is controlled to minimize measurement errors. In some cases, special light arrangements are used to assist vision algorithms [5]. On the other hand, computer vision is also used for inspecting artifacts in a dynamic environment, such as agriculture [6]. However, these are

mostly classification problems such as detecting diseases, insects, and quality of produce, rather than measuring geometric properties. The application concerned in this paper is an automated inspection and assessment system developed to be used in the construction industry. It involves inspecting the correctness of dimensions, angles, and other physical properties of the installation made by trainee workers. In this application, the environment is not controlled as in a manufacturing line, therefore, it is a challenging problem to tackle [7,8].

In the local construction industry, workers are required to go through a training and certification process in order to ensure the quality of building and construction standards. After the training period, workers go through an assessment process for their certification in which they are expected to build a trunking assembly from its technical drawing in a given time period. All the trainees receive the same materials and build the same assembly. The quality of the work produced is then examined by a team of experts manually and visually. The assessment involves checking measurements, alignments, and positions of the assembly, and it is rated based on its compliance with the technical drawing given. Figure 1 shows examples to work produced with alignment issues as well as the manual assessment process. Naturally, it is an arduous and time-consuming process for the training providers. Therefore, a computer vision solution is developed to automate the assessment process in order to speed up.

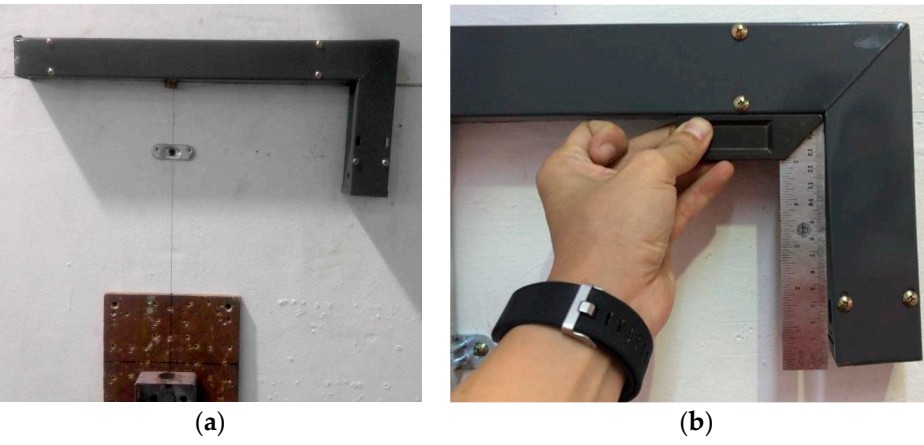

| (**a**) | (**b**) |

**Figure 1.** (**a**) Example of poor workmanship and alignment issues. (**b**) Manual assessment of trainees' work.

Computer vision and image processing techniques in the building and construction industry has long been utilized typically for construction safety, resource tracking and activity monitoring, surveying, inspection, and condition monitoring. For a comprehensive survey on computer vision applications in the building and construction industry, refer to Martinez et al. [3] Smart construction, quality inspection for construction products, and off-site construction are also seen as future research avenues in this field [3]. The research work presented in this paper evaluates the quality of construction products done though it's aimed as an assessment tool for construction workers' training and qualification.

In our application, a vision system inspects the work produced by a trainee in terms of the geometric measures, orientation, and position. It produces a report highlighting discrepancies from the specifications. The primary object feature used for the analysis is the object corners. Consequently, the performance of the system is highly dependent on the accurate detection of the corner points of the objects. Corner detection is widely used in image processing and there are numerous algorithms available in the literature most popular ones being FAST [9], SUSAN [10], and Harris corner detection algorithms [11]. These algorithms are pixel-based operations and they often return redundant or false corner points. In this study, we experimented with FAST, Harris, and Minimum Eigen value corner detection algorithms that are available in Matlab image processing libraries.

We observed that Minimum Eigen value performed well. However, it is computationally costly to filter out noisy detections in order to find corners that belong to the reference points of the objects for our analysis. On the other hand, convolutional networks (CNN) and evolutionary techniques are gaining popularity as corner detectors providing competitive results, especially with noisy images [12]. Recently, deep learning architectures employing many layers of CNNs have become very popular in computer vision due to their remarkable performance. Our experiments showed that deep learning can identify object corners effectively, however, it is within a bounded box [13]. Using a hybrid approach that applies a pixel-based filter, such as Minimum Eigen value algorithm, and further filtering the results with the backing of deep learning output, we are able to detect corners of the objects that we are interested in very accurately. This enabled making reliable geometric measurements and inspections in relatively less controlled environmental conditions.

Recently, we began to see deep learning techniques applied in the building and construction industry. For instance, the presence of workers, equipment, and materials are detected from camera images on sites to improve safety and productivity [14]. Advances in machine learning present many opportunities to be exploited in this industry, such as site supervision, cost prediction, intelligent maintenance, and many more. A detailed survey of various machine learning algorithms, ranging from simple to highly complex techniques, and their applications are presented in [15,16]. Another field where the latest deep learning architectures are dominantly utilized is medicine. Medical images are obtained through various technologies such as ultrasound, magnetic resonance, CT-Scan, and so on. Analysis of these images requires expertise, where recently, deep learning algorithms are used in assistance (see, for instance, tracking tissue in ultrasound videos [17], medical image segmentation [18]). A comprehensive survey on the application of deep learning algorithms in medical image analysis can be found in [19]. As mentioned, deep learning algorithms include many layers of CNNs. After training, these layers become very effective in detecting useful features in input images. We now find deep learning is also supporting conventional image processing techniques in such image segmentation [18], image denoising [20], corner detection [21], and edge detection [22]. In this study, our method for the corner detection is hybrid, where we employed deep learning to detect only the objects that we are interested and their corner locations. We then applied a conventional corner detection algorithm within those boundaries to acquire their precise locations.

In the following, an overview of the system, together with the hybrid solution and performance results will be presented. Section 2 presents key building blocks of the system and their performance. Section 3 presents its operation and user interface. Section 4 presents the performance of the system in achieving an accurate assessment of the trainee's work. An earlier version of this study was presented in [13].

## 2. System Overview

At the earlier phase of this study, off-the-shelf components were used for the hardware setup [13]. Currently, vision hardware is upgraded to an industrial camera (EXO 540CU3 by SVS Vistek, Seefeld, Germany) with 50 mm lens and lighting system since project become ready to be used in real application environment. The camera is set at the midline of the wall assembly. The acquired image resolution is $5320 \times 4600$ pixels. The distance between the camera and wall assembly is set as 180 cm. The acquired images have negligible lens distortion. The camera is calibrated using checkerboard images and using Matlab camera calibration tools [23]. The hardware setup and test assemblies used in the experiments are shown in Figure 2. Two black tapes on the wall assembly are used as a reference, in order to measure the height of the assembly from the floor, which is also an assessment component.

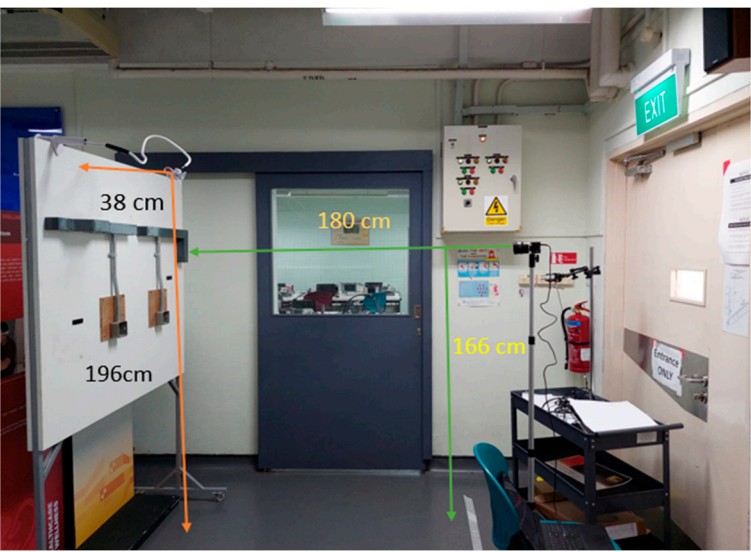

**Figure 2.** Hardware setup and test assemblies used in the experiments.

Matlab is used for the software development. Figure 3 shows an overview of the operations performed in software. After image acquisition, the very first step is to detect the type of wall assembly using a trained deep learning model. This step helps the system to decide which assembly type is subject to analysis so that it can perform computations accordingly. In the next step, object corner points are detected using the hybrid approach that will be described in detail in the following sections. These corner points are used as reference points in the measurement of object dimensions, alignments, angles, and so on. A template of the objects in the wall assembly is available as a CAD drawing, hence, the expected work from the trainee is known prior. The next step of the process is to match detected features with the template and determine all the discrepancies. Furthermore, an examiner can manually mark any points of interest in the image and obtain the measurements for them. Finally, the system generates a report about the inspection results and records it with the candidate's particulars.

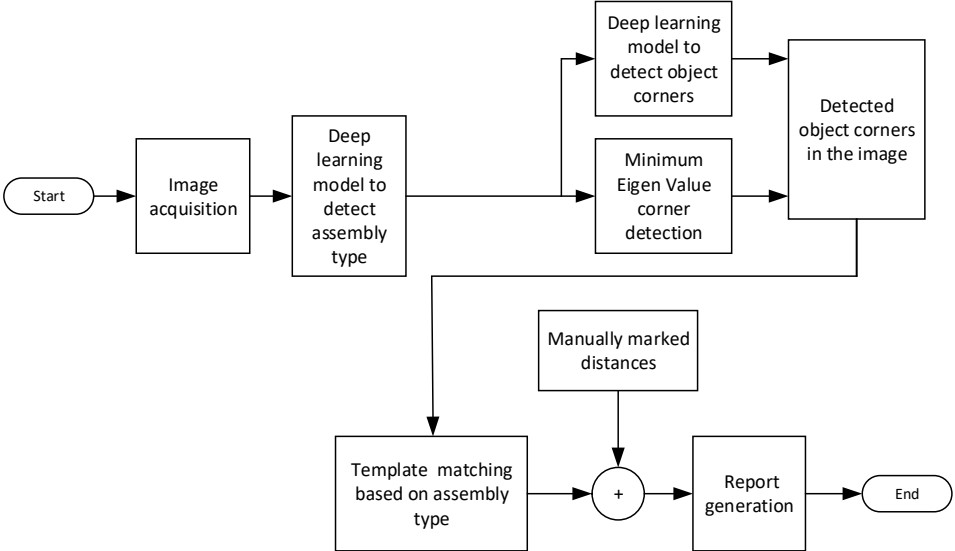

**Figure 3.** A flow chart of the operations.

## 2.1. Detecting Assembly for Inspection

The current system is designed to inspect two different types of electrical trunking assembly as requested by our industry partner. In the remaining, we will use the term

assembly A and B to describe them. Type A assembly is a rather simple L-shaped trunking that has to be installed with the correct dimensions, as shown in Figure 4a. On the other hand, Assembly B tasks involve bending trunking to a certain angle with a correct length and alignment, as shown in Figure 4b. For Type A assembly, the key inspection points are the size, length, and the angle of the trunking, as well as the position and placement of the conduit and switchbox. In addition, for assembly B, the angle of the protruding trunk needs to be measured, which is rather challenging. As mentioned, the first step of the vision system is to identify the type of assembly it is being examined. In this way, we are able to initiate the right inspection algorithm depending on the assembly that appears in the input image. Deep Learning network architecture used in this study is YOLO version 2. YOLO v2 is trained to do classification and bounding box regression at the same time. Additionally, YOLO v2 learns generalizable representations of objects, therefore, it can perform better when applied to new domains or unexpected inputs. There are many well-known meta-architectures in the literature, such as SSD (single shot multibox detector), Faster R-CNN and later versions of YOLO V5 [24]. However, in this study, we are simply concerned with a practical application of a state-of-the-art deep learning network rather than discussing and comparing their performances. For the feature extraction network, we employed Resnet-50 based on our empirical study comparing network and feature extractor architectures. Resnet-50 consists of 50 layers and the pre-trained network can classify images in up to a thousand object categories. The detection sub-network is a small CNN compared to the feature extraction network and is composed of a few convolutional layers and layers specific for YOLO v2.

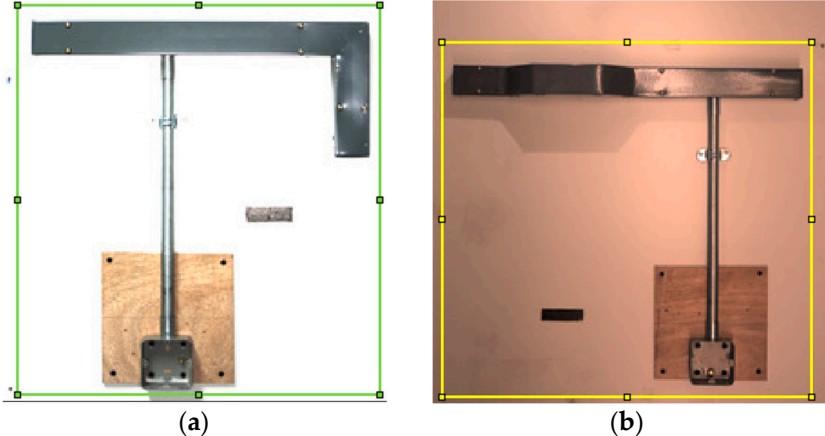

(**a**)　　　　　　　　　　　(**b**)

**Figure 4.** Detected (**a**) Type A assembly and (**b**) Type B assembly.

The training dataset was created by labeling the images using 'Image Labeler' application of Matlab. Test images included samples with different lighting conditions. The training data set enriched further with augmented images with various tilts and rotations, making it a balanced dataset. The number of training images used for Assembly A is 600 and Assembly B is 630. Subsequently, both models were trained and tested.

Intersection over union (IOU) is an evaluation metric used to measure the accuracy of an object detector on a particular dataset. The IOU measures the overlap between the ground truth box and the predicted bounding box. The ground truth box is manually marked boxes during the labeling process of datasets. The predicted bounding box is the bounding box returned by the trained model. IOU is simply the ratio of intersection over the union of these two bounding boxes where 100% result implies perfect overlap between ground truth and predicted [25]. An object detector is considered good if the IOU value is above 50%.

Intersection over union computations are applied in testing both Assembly A and B detectors, as shown in Figure 4, where the green box represents the ground truth box whereas the yellow box represents the predicted bounding box. The IOU values of 20 test

images for both assemblies are shown in Table 1. As seen from the table, the average IOU scores were 0.933 and 0.940, indicating that both assembly detectors were demonstrating superior performance.

**Table 1.** IOU results for the test images.

| Image | Assembly A | Assembly B |
|:-----:|:----------:|:----------:|
| 1 | 0.9462 | 0.9623 |
| 2 | 0.9515 | 0.9442 |
| 3 | 0.9751 | 0.9397 |
| 4 | 0.9420 | 0.9221 |
| 5 | 0.9466 | 0.9487 |
| 6 | 0.9467 | 0.9399 |
| 7 | 0.9466 | 0.9512 |
| 8 | 0.9464 | 0.9444 |
| 9 | 0.9514 | 0.9626 |
| 10 | 0.9468 | 0.9182 |
| 11 | 0.9025 | 0.9446 |
| 12 | 0.9332 | 0.9579 |
| 13 | 0.9378 | 0.9359 |
| 14 | 0.9378 | 0.9272 |
| 15 | 0.9244 | 0.9352 |
| 16 | 0.9030 | 0.9397 |
| 17 | 0.8985 | 0.9308 |
| 18 | 0.9369 | 0.9399 |
| 19 | 0.8928 | 0.9270 |
| 20 | 0.9013 | 0.9270 |
| Average | 0.933 | 0.940 |

*2.2. Detecting Corner Points*

Detection of the reference points of the objects in the assembly and their measurements is the next step in the assessment process. Our method is primarily built upon the accurate detection of corner points of the objects. As these are manmade objects, corner points are the most prominent features. There is a vast literature on corner detection, which is one of the fundamental image processing algorithms. We have experimented with three well-known algorithms, namely Minimum Eigen, Harris, and FAST. We observed that under good lighting conditions all the algorithms are able to detect corner points with test images for both assembly types. However, under poor lighting conditions, the Minimum Eigen algorithm captured most of the corner points that we would like to detect. In particular, assembly B images captured under a single light source due to our inspection schemes and illumination, in this case, are not ideal. Output from these three algorithms are shown in Figure 4 for comparison.

Although the Minimum Eigen algorithm is capable of detecting the corner points of the objects that we are interested in, there are still redundancies and errors present, as seen from Figure 5a. The quality of the image, including its resolutions and lighting conditions also contribute to these errors. On the other hand, recent developments in computer vision demonstrate that deep learning algorithms are very effective in object detection, particularly in a natural environment with much background clutter. In the next phase of the corner detection process, a trained deep learning network is used for detecting key reference corner points of the objects in the image. The bounding boxes generated by the trained network as the region of interest (ROI) are not as precise as the pixel-level detection of corner points. Therefore, we cannot use them directly for measuring object features since they will not be accurate. However, using the ROI determined by the network, we are able to filter out redundant/ erroneous corner points detected by the Minimum Eigen algorithm. This hybrid approach yielded precise detection of object corner points in the image, as it can be seen in Figure 6.

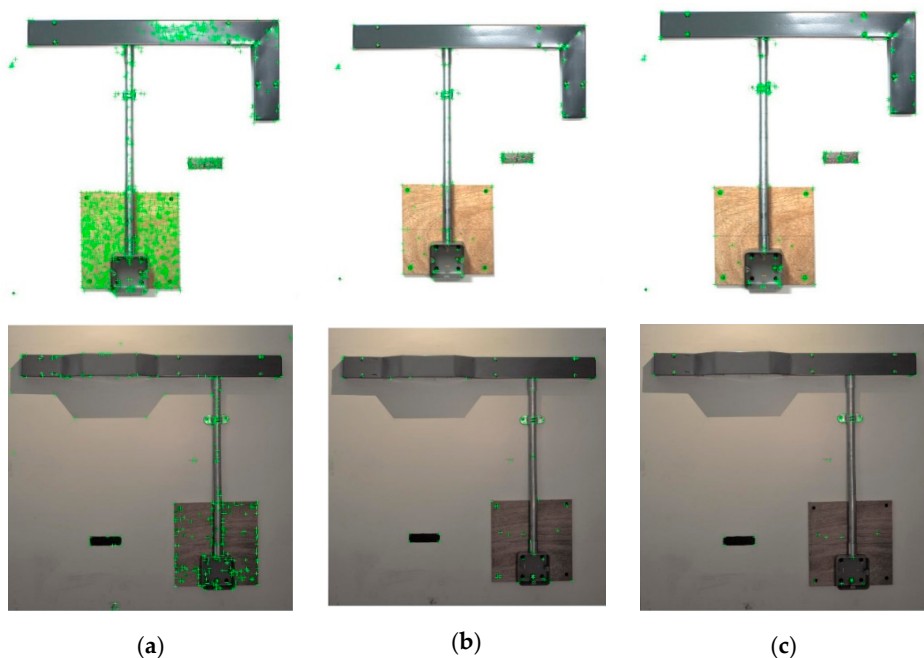

(**a**)          (**b**)          (**c**)

**Figure 5.** Experiments with corner detection algorithms for both assembly types. (**a**) Minimum Eigen, (**b**) Harris, and (**c**) FAST. Green marks indicate raw corner points returned from algorithms. Parameters such as minimum accepted quality of corners set as 0.01 and filter size as 5 for all the algorithms.

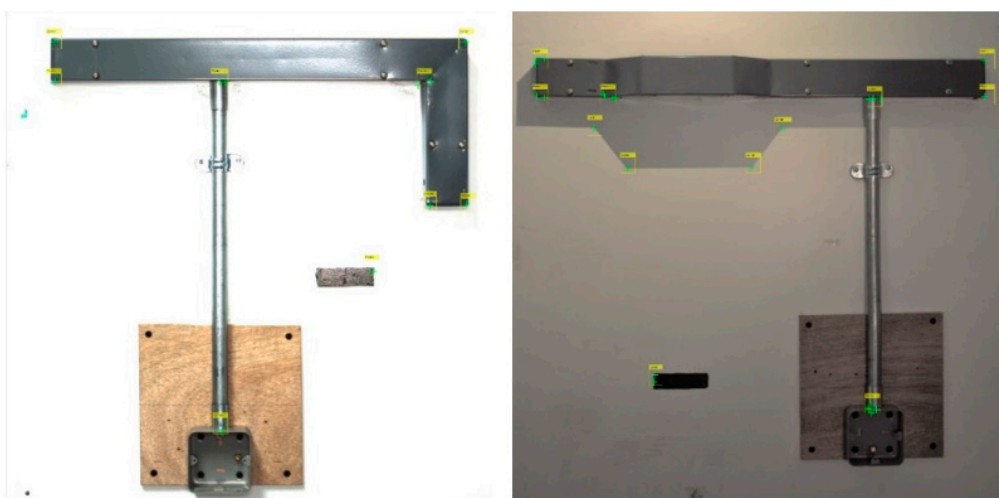

**Figure 6.** Error filtering using trained deep learning networks. Corners detected after error filtering and marked.

### 2.3. Deep Learning for Detecting Corner Points

We consider using a pre-trained network when developing models for detecting the wall assemblies and corner points of objects. A model's performance heavily relies on the pre-trained network's ability to extract dominant and needed features from the datasets during its training. Instead of developing and training the network from scratch, we use pre-trained networks that are relatively well-known with their good performance. We tested and compared the performance of pre-trained networks Resnet-18, Resnet-50, and Resnet-101. For this experiment, we utilized the deep learning algorithm architecture YOLOV2. Training datasets used in this experiment include 879 images for assembly A and 700 for assembly B. Data augmentation is used to add more variety to the training data from the labeled images. In this case, flipped and rotated images are added to the training set. Key training parameters are set as follows. Image size is $224 \times 224 \times 3$. The

Relu activation function is used in all the networks experimented. The stochastic gradient descent with momentum (SGDM) optimizer is used to solve the training network since it provides a faster conversion. Mini-batch size of 10 and an initial learning rate of 0.001 are used. Training takes a longer time if the learning rate is low. On the other hand, training may achieve poor results if the learning rate is too high. Lastly, the number of epochs to complete the training session was 100. We did not use larger values to evade over the fitting problem.

Table 2 shows the performance of each pre-trained network model for both assembly A and B during training. After observing precision and recall curve graphs, we observed that Resnet-50 well suited pre-trained network to employ since it yields a higher area under the curve and delivers high precision (for the simplicity of presentation, we did not include these graphs). Resnet-101 had higher average precision though it is a complex network with a longer training time. Resnet-18 was faster to train though precision and recall curve graph showed a smaller area under the curve.

**Table 2.** Performance comparison of pre-trained networks during training.

|  | Pre-Trained Networks | Average Precision (%) | Elapsed Time (min) |
|---|---|---|---|
| Assembly-A | Resnet-18 | 96.9 | 105 |
|  | Resnet-50 | 96.2 | 386 |
|  | Resnet-101 | 97 | 515 |
| Assembly-B | Resnet-18 | 92.8 | 88 |
|  | Resnet-50 | 92.9 | 275 |
|  | Resnet-101 | 96.7 | 432 |

Considering that vision-based inspection system is going to operate under dynamic environmental conditions rather than a controlled one, we further evaluated the accuracy and consistency of the trained network. A set of 20 test images with different lighting conditions and camera angles were used. The accuracy is defined as the ratio of accurately detected corner points over actual number of object corner points in the image. Table 3 tabulates these test results. Majority of test results lie above 93% accuracy, however, there were also some false detections as well as relatively poor accuracies as low as 75%, mainly due to extremely poor lighting and image quality of the selected test images.

**Table 3.** Network performance on test images.

| Image | Corners Expected | Corners Detected | Corners Missed | Accuracy | False Detection |
|---|---|---|---|---|---|
| 1 | 12 | 12 | 0 | 100% | 1 |
| 2 | 12 | 12 | 0 | 100% | 1 |
| 3 | 12 | 12 | 0 | 100% | 1 |
| 4 | 12 | 12 | 0 | 100% | 3 |
| 5 | 12 | 12 | 0 | 100% | 1 |
| 6 | 12 | 12 | 0 | 100% | 1 |
| 7 | 12 | 12 | 0 | 100% | 1 |
| 8 | 12 | 12 | 0 | 100% | 0 |
| 9 | 12 | 12 | 0 | 100% | 1 |
| 10 | 12 | 12 | 0 | 100% | 2 |
| 11 | 12 | 11 | 1 | 91.7% | 0 |
| 12 | 12 | 10 | 2 | 83.3% | 0 |
| 13 | 12 | 11 | 1 | 91.7% | 0 |
| 14 | 12 | 10 | 2 | 83.3% | 1 |
| 15 | 12 | 9 | 3 | 75% | 1 |
| 16 | 12 | 8 | 4 | 66.7% | 0 |
| 17 | 12 | 10 | 2 | 83.3% | 0 |
| 18 | 12 | 10 | 2 | 83.3% | 0 |
| 19 | 12 | 12 | 0 | 100% | 0 |
| 20 | 12 | 12 | 0 | 100% | 0 |
| Average |  |  |  | 92.2% |  |

### 3. Template Matching and Inspection of the Wall Assembly

The above-mentioned hybrid approach provided a robust detection of corner points that are needed for our analysis. Nevertheless, there may still be a few remnant stray corner points. As the technical drawing of the wall assembly is known prior, in the final step of the analysis, a template matching method is used to establish the most suitable corner points to represent the corners of the objects in the assembly. The template is constructed starting from the top left corner point using geometric methods. The selection of valid corner points is made by accepting the corner points that are closest to the template's corner points, thereby completing the corner point selection process. Figure 7a shows the best fitting template, and Figure 7b the final reference corner points extracted. Once these reference points are obtained, measurements for assessment are done using basic geometric techniques. Finally, a report, as an excel spreadsheet, is created where the result of the analysis, listed as shown in Figure 7c.

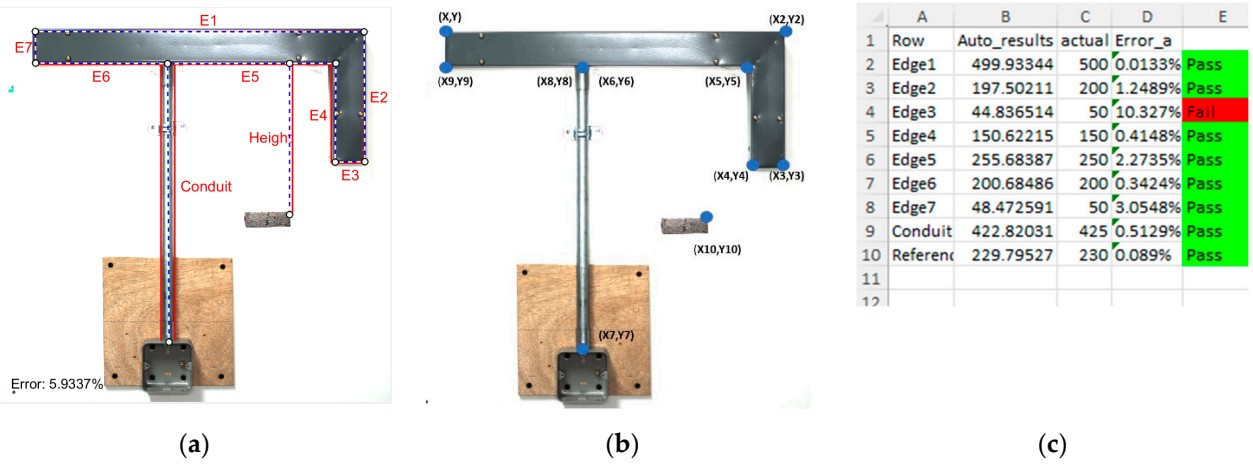

| | (a) | (b) | (c) |

**Figure 7.** Template matching for Assembly A. (**a**) Template of the object fitted on the image. Edges labeled 1 to 6 and conduit length are some of the key measurements (**b**) Detected reference corner points illustrated on the image for better visualization. (**c**) A screen capture of the final report generated at the end of assessment.

A key requirement for Assembly type B was the measurement of the protruding angle of the trunking. The vision hardware unit is positioned in a fixed distance from the wall assemblies and needs to remain fixed for the sake of calibration accuracy. However, hardware unit can be shifted laterally to capture images. A light source positioned above the camera lens with a fixed angle is used for projecting a shadow of the trunking as shown in Figure 8a. The angles of protruding part of the trunking are then measured from the shadow of the object rather than the object itself, as shown in Figure 8b,c. Given the position of the light source, angles of the protruding part of the trunking on the shadow are calculated as 130 degrees using basic geometric transformations. Hence, a correctly fabricated bent trunking measurement should be measured as 130 degrees; a deviation from this indicates that there are flaws in the fabrication. Similarly, an Excel report is created providing the analysis results, as shown in Figure 8d. For both assembly types, after object dimensions are analyzed, a best fitting template of the actual objects in the assembly is used to illustrate user the outcome of the analysis visually. As mentioned, the template matching algorithm starts a search process to find the best fit starting from the top left-most corner point detected. A comparison of the template with the analyzed objects from the image enables the user to see the amount of deviation. If this deviation is very large, the examiner may have to adjust the hardware before taking another image. There is still a possibility of either the minimum Eigen corner detector or the deep learning model may not be able to detect corner points or make false detections. This may occasionally result in failures in the analysis. In order to provide a workaround for such cases, we also provided manual

assessment options where users can mark points of interest over the image manually and obtain the measurement of distances or angles.

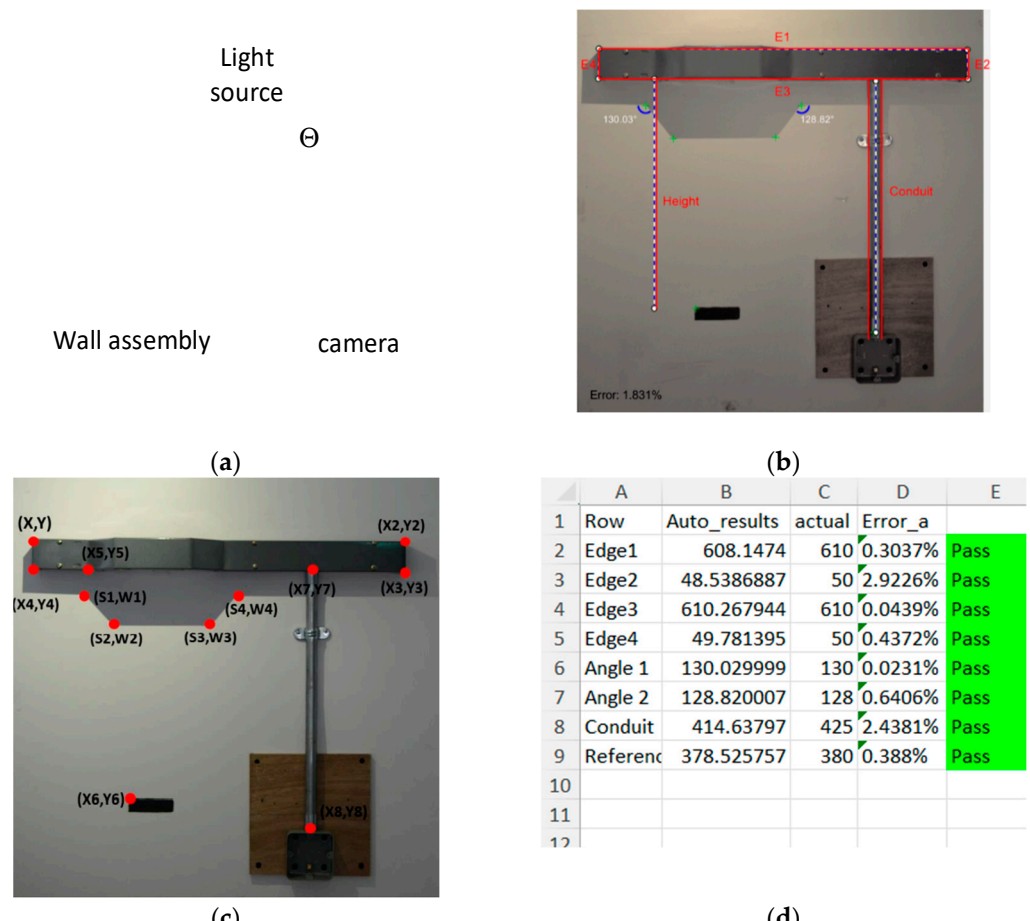

**Figure 8.** Template matching for Assembly B. (**a**) A block diagram showing camera and light arrangement. (**b**) Template of the object fitted on the image. Edges labeled 1 to 4, conduit length and two angles of the bent trunking are the key measurement points (**c**) Detected reference corner points illustrated on the image for better visualization. (**d**) A screen capture of the final report generated at the end of assessment.

*Graphical User Interface*

In this session, the user interface of the system will be described. After selecting the type of assembly to inspect, the user is prompted with a menu, as shown in Figure 9a. The image on the left is for assembly A and the one on the right is for assembly B. The analyze image button will open the file explorer to upload the captured image from the assembly made by the trainee. Inspection will be run automatically and the results will be displayed as shown in Figure 9b. Considering possible detection errors for each assembly, a list of manual marking options are given, as shown in the user menu. In this way, the user can still manually obtain measurements if it is missed during auto-inspection. Users can also mark other areas of interest to get their measurement as shown in Figure 9b. Finally, the "create report" and "save image" button brings up a pop-up window to enter trainee particulars and examiners' comments. The resulting report is saved as an excel file together with the image analyzed. A video demonstration of the final system and its usage is provided in [26].

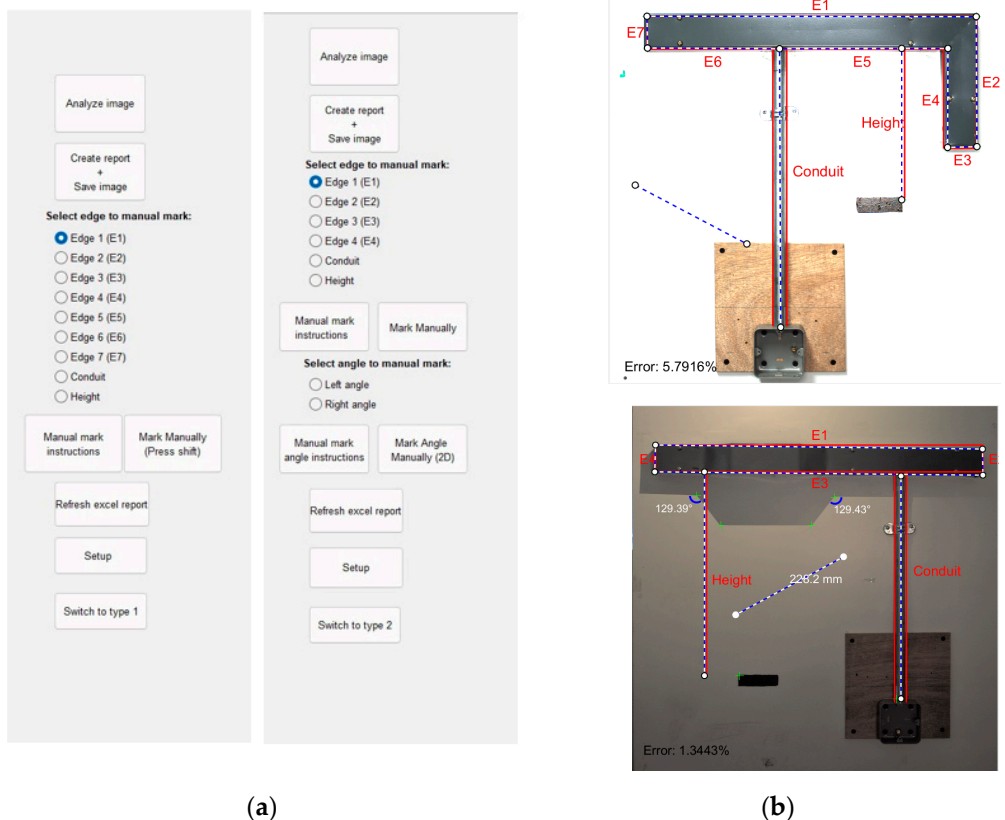

**Figure 9.** (**a**) User interface for the inspection system, and (**b**) manual measurement of points of interest by the user.

## 4. Performance

In order to evaluate the performance of our system, we used a test setup that is fabricated accurately, as indicated in the technical drawings given to trainees, as shown in Figure 1. Using the hardware setup, we have taken images from the fixed point, however, under slightly varying lighting conditions. A total of 30 images were used for testing each assembly. We obtained the measurement errors for each assessment component in all the images. For the purpose of illustration, we tabulated measurement results in Table 4 using only five images taken under normal lighting conditions for Assembly A. These results compare the actual properties of the correctly made assembly versus what is measured by the system. The difference between them consequently implies the measurement error of our system. Table 4 lists all the measurements for each test image and average errors. In most of the cases, the average error lay below 1%, which implies a negligible measurement error. However, it is also noticeable that the average percentage of error for some edges (Edges 3 and 7) were relatively higher than the other edges. These larger errors were due to the inaccuracies generated during template matching, particularly caused by shadows of the objects. However, these two measurements are trivial since they correspond to the width of the trunking, which is standard. Similar observations have been made for Assembly B. Table 5 lists the measurement results versus actual values for a selected number of images taken under normal lighting conditions for the purpose of illustration. The critical measurement point in this assembly is the protruding angle of the trunking. Results show a minor 0.2% average measurement error for them. It is important to note that here we provide average error as a simple performance indicator. Nevertheless, as required by our industry partner, in any of the measurement points, more than a 5% error is considered a non-negligible system error. In all the test images taken under normal conditions, we did not observe a major system error.

**Table 4.** Average measurement (in millimeters) error for test images (Assembly A).

| Feature | Image 1 | Image 2 | Image 3 | Image 4 | Image 5 | Actual | Average Measurement Error (%) |
|---------|---------|---------|---------|---------|---------|--------|-------------------------------|
| Edge 1 | 499.9 | 499.9 | 499.8 | 499.4 | 499.6 | 500 | 0.06 |
| Edge 2 | 197.5 | 197.4 | 196.8 | 196.4 | 197 | 200 | 1.49 |
| Edge 3 | 44.8 | 45 | 45.7 | 45.5 | 44.5 | 50 | 9.8 |
| Edge 4 | 150.6 | 150.2 | 149 | 149.3 | 150 | 150 | 0.3 |
| Edge 5 | 255.7 | 255.9 | 248.4 | 247.7 | 255.6 | 250 | 1.69 |
| Edge 6 | 200.7 | 200.6 | 200.2 | 200.5 | 200.3 | 200 | 0.23 |
| Edge 7 | 48.5 | 48 | 48.5 | 48.8 | 48.3 | 50 | 3.16 |
| Conduit | 422.8 | 423.1 | 422.5 | 422.2 | 422.2 | 425 | 0.57 |
| Height reference | 229.8 | 230 | 228.3 | 227.9 | 229.4 | 230 | 0.4 |

**Table 5.** Average measurement (in millimeters) error for test images (Assembly B).

| Feature | Image 1 | Image 2 | Image 3 | Image 4 | Image 5 | Actual | Average Measurement Error (%) |
|---------|---------|---------|---------|---------|---------|--------|-------------------------------|
| Edge 1 | 608.1 | 608.6 | 608.7 | 609 | 610.7 | 610 | 0.21 |
| Edge 2 | 48.5 | 48.6 | 48.6 | 48.7 | 48.8 | 50 | 2.72 |
| Edge 3 | 610.3 | 610.5 | 610.5 | 610.9 | 612.6 | 610 | 0.16 |
| Edge 4 | 49.8 | 50 | 49.8 | 49.9 | 50.2 | 50 | 0.28 |
| Angle 1 | 130 | 129.7 | 129.7 | 129.6 | 129.7 | 130 | 0.20 |
| Angle 2 | 128.8 | 129 | 129.1 | 129 | 129.2 | 129 | 0.08 |
| Conduit | 414.6 | 414.5 | 414.9 | 415.2 | 415.7 | 425 | 2.36 |
| Height reference | 378.5 | 378.6 | 378.4 | 378 | 379 | 380 | 0.39 |
| Edge 1 | 608.1 | 608.6 | 608.7 | 609 | 610.7 | 610 | 0.21 |

In some odd cases, we observed large errors (i.e., more than 5%). In a closer look into the root cause of these errors, we observed that there were slight deviations between detected and actual reference points, as illustrated with examples in Figure 10. There were two major causes for this error: rotation and/or illumination. It was expected that both test assemblies were not tilted, however, a minor tilt can happen while capturing the image by the user if the camera is not fixed properly. Another possibility is that the assembly produced by the trainee was tilted, which should have been made parallel to the floor. In this case, the reference corner points obtained from the images would cause an inaccurate template match and would increase the errors in the measurements. Based on our experiments, we observed that if the measurement errors were more than 5%, it was due to these external factors or the wall assembly done very badly by the trainee. We did not include a rotation function to fit the template if the assembly was tilted in making since such assembly is considered failed. Nevertheless, other parameters can still be measured if needed by correcting tilt using an image editor before the analysis. The other source of error was due to the presence of shadows. They appear when there is an uneven distribution of light sources over the assembly during image capture. When the difference in light intensity between the shadow and the object itself is not wide enough, the corner detector algorithm will capture the corner point in the shadow, instead of the actual object. Consequently, template matching was not able to correct this error and the shadow's corner was used for measurements. This also produced inaccurate results, however, errors due to shadows can be minimized with better lighting arrangement in hardware.

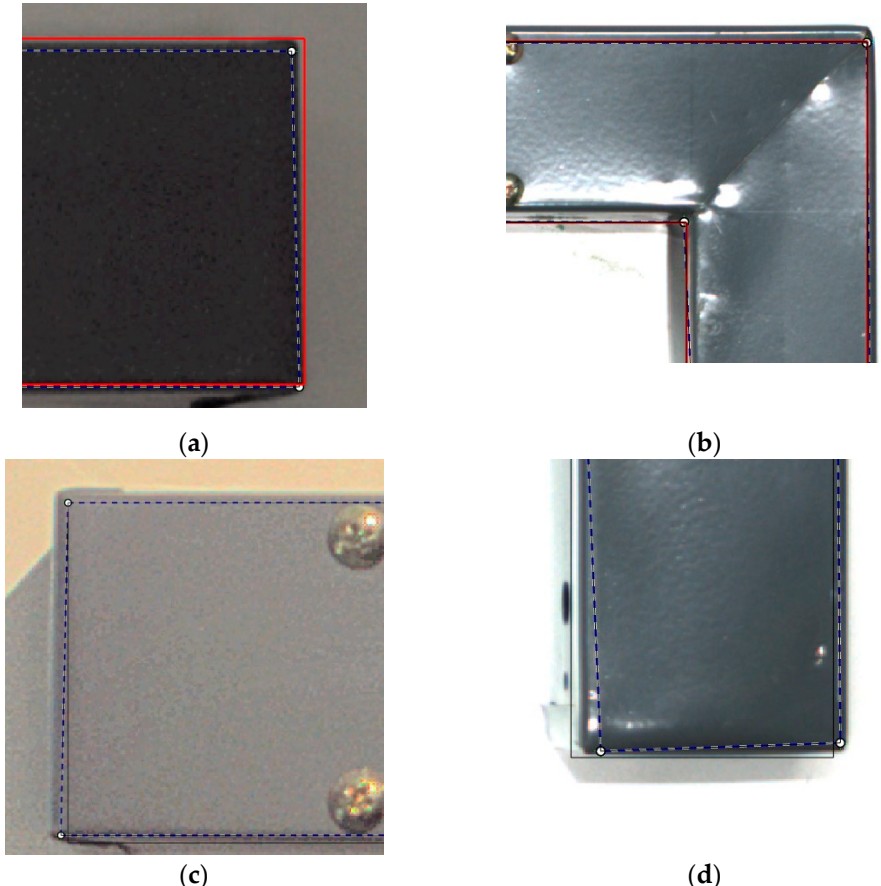

**Figure 10.** Examples of measurement errors (**a**,**b**). The template drawn deviates from the actual object (**c**,**d**) with more severe deviations due to errors caused by image and light quality.

## 5. Conclusions

This paper described a computer vision-based inspection system for automated assessment of work done by trainees in the construction industry for their certification. The primary objective of the study was to minimize the mundane manual/visual assessment process and save person hours. However, in order for the computer vision system to replace the manual assessment method, it should make geometric measurements accurately. A key feature used for this purpose was the corner points of the objects. Accurate detection of the corner points contributes to the system's accuracy and performance significantly. Intuitively, object corner points in the image should be detected within a few pixel accuracy. The minimum Eigen Value corner detection algorithm is used for this purpose. In this application, the environment is not fully controlled when compared to, for instance, a quality control application in manufacturing. Therefore, the corner detection algorithm may return many redundant points. In order to filter them out and identify the ones that belong to the objects analyzed, we utilized deep learning algorithms. This hybrid approach provided accurate detection of object corners. Using the template matching method, measurements of detected objects such as dimensions, alignments, and angles are compared with the template obtained from the technical drawings. Our experimental results revealed that the vision system has minor measurement errors under good lighting conditions. In summary, the computer vision-based solution reduced the time spent significantly compared to manual inspection. Moreover, the physical presence of experts at the assessment ground is not required as image capturing and processing can be done remotely. An interesting contribution of this study is the hybrid corner detection approach which enables computer vision techniques to analyze and evaluate manufactured objects in dynamic environments.

It provides avenues to explore other quality control and inspection problems in rather less controlled environments.

Our future study involves developing a generic system where various assessment configurations can be programmed by the instructors creating a wide range of scenarios that can be used for examining trainees' work and grading them.

**Author Contributions:** Conceptualization, M.F.E. and R.B.W.; methodology, M.F.E. and R.B.W.; software, R.B.W.; validation, M.F.E.; formal analysis, M.F.E. and R.B.W.; investigation, M.F.E.; resources, R.B.W.; datacuration, R.B.W.; writing—original draft preparation, M.F.E. and R.B.W.; writing—review and editing, M.F.E. and R.B.W.; visualization, M.F.E. and R.B.W.; supervision, M.F.E.; project administration, M.F.E.; funding acquisition, M.F.E. All authors have read and agreed to the published version of the manuscript.

**Funding:** This project is sponsored by Singapore Polytechnic under the Technology Harvesting Grant 2020. Project title: Computer vision-based solution to electrical installation and workmanship quality inspection. Grant number 03-11000-36-J723.

**Institutional Review Board Statement:** Not applicable.

**Informed Consent Statement:** Not applicable.

**Data Availability Statement:** The data presented in this study are available on request from the corresponding author.

**Conflicts of Interest:** The authors declare no conflict of interest.

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
