# Peer review of "Computer Vision-Based Inspection System for Worker Training in Build and Construction Industry"

_computers, doi:10.3390/computers11060100_

Round 1

Reviewer 1 Report

Authors proposed a new method of computer vision for assessment of training and assessment of construction works. 

In particular, two different tasks proposed to be done by deep learning, object detection to compare two assemblies and point detection for detecting corner points. In the second method, they utilized traditional methods and deep learning method. 

The application of deep learning for inspection and assessment of construction tasks is really interesting. However, I cannot see the difficulty and challenge of the dataset. From figures, objects are limited and straightforward. I cannot see the rational for using YOLO v2 for detecting this limited image dataset. 

Revisions that should be done before consideration for publication:

- Clear explanation of methods. Figure 2 is really vague and convoluted. This section should be re-write with all details of deep learning model, hyper-parameter tuning, training validation, annotation method, learning strategy, augmentation technique if it used, and so on. In the comparison section, I can see 5 images, is the testing dataset is 5? Average error, where is standard deviation? 

Conclusion is weak, and should be expanded. 

Technically, I see problems in the work. Table 2 for example, is testing images really took more than one hour? 

Citations are not enough to cover the literature. Add articles related to object detection and point detection. 

Add a link for sharing your codes and possibly dataset on a repository. 

Figures such as Figure 6 and 7 must be a table not a screenshot

Author Response

We thank referees for their comments and suggestions. The following are the revisions made accordingly:

-Clear explanation of methods. Figure 2 is really vague and convoluted. This section should be re-write with all details of deep learning model, hyper-parameter tuning, training validation, annotation method, learning strategy, augmentation technique if it used, and so on. In the comparison section, I can see 5 images, is the testing dataset is 5? Average error, where is standard deviation?

Network optimizer, hyper parameter tuning and other relevant details provided. Experiments conducted for 30 images for each assembly. Results for 5 test images are given for the purpose of illustration. Average error is used as performance indicator to simplify tables. In any measurement point, system error should not be above 5% as set by the industry partner. These are clarified in the write up.   

- Conclusion is weak, and should be expanded. 

Goal of this work is better explained in conclusions now and contribution of the study emphasized.

- Technically, I see problems in the work. Table 2 for example, is testing images really took more than one hour? 

These are training time for developing the model not detection time. To avoid confusion we indicated on table caption and in the writing.

-Citations are not enough to cover the literature. Add articles related to object detection and point detection. 

Object detection and corner detection methods are well known in computer vision community. However, we included more references as it deem fit with the revisions.

- Add a link for sharing your codes and possibly dataset on a repository. 

A video showing the various steps of the development and its operation is provided.  Providing code and data is not possible at this stage of the project as it may have commercial value.

-Figures such as Figure 6 and 7 must be a table not a screenshot

The purpose is just to illustrate excel report generated by the system referring to input image shown. Figure captions revised to better explain.

Reviewer 2 Report

The authors presented a vision-based approach for analyzing the results of worker training assessments. An algorithm was developed based on different techniques in image processing and deep learning to detect objects, edges, etc. The presented approach was found to be reasonable. However, it can be noticeable that several aspects of the algorithm were not properly explained such as model optimization, selection of arbitrary algorithm parameters and so on. It is highly recommended to review the said aspects since they are critical in proving the scientific contribution of this work. Besides that, the English writing style and grammar should be significantly improved since there were many parts that were difficult to comprehend due to the style of writing. There structure is also considered as an issue. Here are some specific comments and issues found in the current version of the manuscript:

  1. Abstract: Please recheck this part as there are many awkward usages of English. i.e., L17: “The” system described in this paper.
  2. L31: Most of these applications “are” concerned
  3. L118: Reminder?
  4. L127: Why YOLO v2 and not v3/v4/v5?
  5. L150: Please provide the mathematical equation of IoU to inform readers about this concept
  6. Table 1: I believe this should be situated somewhere in a Results section
  7. Table 1: Are these the results after training the YOLO v2 model? How was the YOLO v2 model trained? I believe this is something critical and should’ve been explained in detail.
  8. I understand that the authors tried to create their own structure for the paper however the current structure was difficult to follow.
  9. L171-172: What were the block size or other arbitrary parameters used in using the three algorithms? Were there any optimizations performed? All of these should be properly stated in the paper.
  10. L184: Were the errors quantified in any way?
  11. L207-209: How were these networks optimized?
  12. What was the resolution of the images used? Is there any setup in image acquisition? If Section 3 explains about this, there should be a dedicated section for the image acquisition setup, which should come earlier in the text rather than at the near end.
  13. L316-317: It was mentioned that minor tilts affected the results. Is there any way to have a reference object that can be used to reduce the error due to tilts? If yes, then it is better to mention about this.
  14. Are the materials used for the certification tests uniformed? If yes, this should also be mentioned.
  15. I feel that the title should also conform more on the topic presented in this work. The current title seems to be too general/broad.
  16. It is also confusing whether the goal of this work was high accuracy or fast processing. It has to be clarified in the conclusion.

Author Response

-Abstract: Please recheck this part as there are many awkward usages of English. i.e., L17: “The” system described in this paper.

Abstract revised accordingly.

-L31: Most of these applications “are” concerned

Corrected.

-L118: Reminder?

Corrected.

-L127: Why YOLO v2 and not v3/v4/v5?

YOLO V2 heavier and slower network though it has high detection performance. The main advantage of later version of YOLO are being light and fast. In our application, a real time detection speed is not vital since we are not processing video stream but image. Nevertheless, this is clarified in the writing and a reference to YOLO comparison is provided.   

-L150: Please provide the mathematical equation of IoU to inform readers about this concept

Revised writing to better define IOU. To simplify writing we provided a reference (reference [16]) for more information and equation on this topic.

-Table 1: I believe this should be situated somewhere in a Results section

-Table 1: Are these the results after training the YOLO v2 model? How was the YOLO v2 model trained? I believe this is something critical and should’ve been explained in detail.

-I understand that the authors tried to create their own structure for the paper however the current structure was difficult to follow.

Paper shows development of the system its components. In section 2, these components described and their performance evaluated individually.  Section 3, presents the overall system its performance in achieving assessment. We intend to stick to the same structure however we added a clarification at the introduction for reader’s convenience. More details also provided for network training.

-L171-172: What were the block size or other arbitrary parameters used in using the three algorithms? Were there any optimizations performed? All of these should be properly stated in the paper.

These parameters are now shown in Figure caption.

-L184: Were the errors quantified in any way?

This is omitted since minEigen was clear winner based on output images particularly under low light conditions.  

-L207-209: How were these networks optimized?

Network optimizer, hyper parameter tuning and other relevant details provided. Experiments conducted for 30 images for each assembly. Results for 5 test images are given for the purpose of illustration. Average error is used as performance indicator to simplify tables. In any measurement point, system error should not be above 5% as set by the industry partner. These are clarified in the write up.

-What was the resolution of the images used? Is there any setup in image acquisition? If Section 3 explains about this, there should be a dedicated section for the image acquisition setup, which should come earlier in the text rather than at the near end.

These details are now provided at the beginning of section 2.

-L316-317: It was mentioned that minor tilts affected the results. Is there any way to have a reference object that can be used to reduce the error due to tilts? If yes, then it is better to mention about this.

This is a minor error and can be corrected by simple image rotation if the root cause is the camera alignment during image acquisition. Therefore no extra effort put for handling this error. 

-Are the materials used for the certification tests uniformed? If yes, this should also be mentioned.

This is a good point and it is clarified in the introduction.

-I feel that the title should also conform more on the topic presented in this work. The current title seems to be too general/broad.

Title revised as “Computer vision based inspection system for worker training in build and construction industry” making it more focused.

-It is also confusing whether the goal of this work was high accuracy or fast processing. It has to be clarified in the conclusion.

Goal of this work is better explained in conclusions now and contribution of the study emphasized.

Reviewer 3 Report

The topic of this manuscript is interesting and significant. The authors would like to apply CV techniques for automatic task assessment at the construction sites. Here are some issues and suggestions from my view.

  • It is suggested to clarify the exact latest object detection technique in the abstract. Besides, the abstract is a bit longer than usual.
  • L28, in construction industry → in the construction industry
  • L56, what is the meaning of “Materials used during assessments are uniform”?
  • In the construction sector, the quality assessment is carried out by comparing the as-built and as-planned buildings. It not only contains geometric visual measurements, alignments, and positions, it also involves materials, invisible steel rebars, etc. This research seems to propose an inspection system for only steel structures assembling.
  • L93, if prior experiments have been conducted before, the authors should add associated references.
  • L147, YOLO has been updated to version 5. Version 2 is not the latest technique.
  • L183, no need to provide the IOU results of test images cause the readers could not examine or reconduct such an experiment.
  • L193, the authors may try data augmentation when facing poor illumination. Besides, it is strange that the corners of shadow could not been detected using these algorithms.
  • L218, in figure 6, the shadow corners could be detected perfectly. There must be some certain reasons.
  • L256, why two assembly-A
  • The statement of deep learning algorithms for corner detection is not clear that the architecture is almost same to ResNet? But the task is totally different. ResNet is the backbone to extract features for classification but how to locate the corners?
  • L273, the figure 7 is not academic.

Author Response

We thank referees for their comments and suggestions. The following lists the revisions or answers to their queries accordingly:

  • It is suggested to clarify the exact latest object detection technique in the abstract. Besides, the abstract is a bit longer than usual.

We rather not to revise anymore since the abstract explains the purpose of the study methodically.

  • L28, in construction industry → in the construction industry

Minor language errors are now corrected

  • L56, what is the meaning of “Materials used during assessments are uniform”?

Reworded as “All the trainees receive the same materials and build the same assembly.”   

  • In the construction sector, the quality assessment is carried out by comparing the as-built and as-planned buildings. It not only contains geometric visual measurements, alignments, and positions, it also involves materials, invisible steel rebars, etc. This research seems to propose an inspection system for only steel structures assembling.

This system is designed only for inspection of trainee work for their qualification.

  • L93, if prior experiments have been conducted before, the authors should add associated references.

Reference [13] provided.

  • L147, YOLO has been updated to version 5. Version 2 is not the latest technique.

 In this study, we are primarily concerned with a practical application of a state-of-the-art deep learning network rather than discussing and comparing their performances. This is also mentioned in the paper.

  • L183, no need to provide the IOU results of test images cause the readers could not examine or reconduct such an experiment.

The bounded boxes, delivered from deep learning algorithm, are used for filtering out redundant corner points therefore  IOU results provided as an factor corner detection performance hence we intend to keep those tables.   

  • L193, the authors may try data augmentation when facing poor illumination. Besides, it is strange that the corners of shadow could not been detected using these algorithms.

This statement refers for accurate measurements of objects directly illumination should be done uniformly as shown in image on the left of Figure 6 else shadows will impact the measurement accuracy

  • L218, in figure 6, the shadow corners could be detected perfectly. There must be some certain reasons.

For assembly B we deliberately use shadow so that we can measure protruding trunking angles.   

These points reworded to be better explain in text.

  • L256, why two assembly-A

This is the requirement of our industry partner.

  • The statement of deep learning algorithms for corner detection is not clear that the architecture is almost same to ResNet? But the task is totally different. ResNet is the backbone to extract features for classification but how to locate the corners?

Minimum Eigen algorithm is used for pixel level localisation of the corners within the bounding boxes returned by deep learning.  Resnet is employed for feature detection needed for deep-learning. These are explained in the paper.

  • L273, the figure 7 is not academic.

I presume this refers to screen capture of excel output. The purpose is just to illustrate report generated by the system referring to input image shown. Figure caption was revised to better explain this.

Round 2

Reviewer 1 Report

Thank you for your new revision. However, the new version couldn't address my concerns. I couldn't find provided details also in the answers by the authors: "Network optimizer, hyper parameter tuning and other relevant details provided."

I also found more errors:

"Deep Learning network architecture used in this study is YOLO version 2. Other meta-architectures that are light weight and fast such as SSD (Single Shot Multibox Detector), Faster R-CNN were also experimented but not considered since in our application accuracy is more imperative than the real-time object detection performance. Same goes to later versions of YOLO as they are mainly lighter and faster [15]. YOLO v2 is trained to do classification and bounding box regression at the same time. Additionally, YOLO v2 learns generalizable representations of objects therefore it can perform better when applied to new domains or unexpected inputs."

This sentence is completely wrong. YOLO is an open-source object detection system. It can recognize objects on a single image or a video stream RAPIDLY. SSD detects objects with higher precision in a single forward pass computing feature map.

Therefore, you must get better results from SSD than YOLO in theory. 

Author Response

We thank referees for their comments and suggestions. The following lists the revisions or answers to their queries accordingly:

Thank you for your new revision. However, the new version couldn't address my concerns. I couldn't find provided details also in the answers by the authors: "Network optimizer, hyper parameter tuning and other relevant details provided."  

Lines 229 to 237 lists down the key parameters concerning training such as image size, number of epoch, optimizer, batch size, learning rate, optimizer used, etc.

I also found more errors:

"Deep Learning network architecture used in this study is YOLO version 2. Other meta-architectures that are light weight and fast such as SSD (Single Shot Multibox Detector), Faster R-CNN were also experimented but not considered since in our application accuracy is more imperative than the real-time object detection performance. Same goes to later versions of YOLO as they are mainly lighter and faster [15]. YOLO v2 is trained to do classification and bounding box regression at the same time. Additionally, YOLO v2 learns generalizable representations of objects therefore it can perform better when applied to new domains or unexpected inputs."

This sentence is completely wrong. YOLO is an open-source object detection system. It can recognize objects on a single image or a video stream RAPIDLY. SSD detects objects with higher precision in a single forward pass computing feature map.

Therefore, you must get better results from SSD than YOLO in theory. 

Above sentence revised to avoid misunderstanding (line 148-152). In this study, we are concerned with a practical application of a state-of-the-art deep learning network rather than discussing and comparing their performances.  

Reviewer 2 Report

The authors were able to properly respond to the issues raised in the first version. The new figures were also very helpful to understand the methodologies presented in this paper.

  1. Just please take note of Figure 4 since it might have been misaligned after submission.
  2. The Figure 2 captions seem to be wrong too

Author Response

We thank referees for their comments and suggestions. The following lists the revisions or answers to their queries accordingly:

  • Just please take note of Figure 4 since it might have been misaligned after submission.

Figure 4 placed into a table now. This should prevent misalignment

  • The Figure 2 captions seem to be wrong too

Thanks for capturing this. It is corrected now.

Reviewer 3 Report

Several comments have not be addressed properly.

Author Response

We thank referees for their comments and suggestions. The following revisions made accordingly:

We  went through the text made minor corrections in typo, formatting as well as rewording some sentences to better express the ideas. These changes can be seen with MS-Word tracking feature. Furthermore, we expand the introduction giving more examples of deep learning application in various fields (lines 110 -130).

Round 3

Reviewer 1 Report

The current version of the manuscript is better than the previous versions. Authors should emphasize more the application of Deep learning which I see in the new version, became better. 

Citation of similar studies (application of deep learning is a better way to increase your work visibility). Add them to your manuscript, for example, in two recent studies, authors used vision systems and the location of points similar to your idea in a different application. 

Mozaffari, M. Hamed, Noriko Yamane, and Won-Sook Lee. "Deep Learning for Automatic Tracking of Tongue Surface in Real-time Ultrasound Videos, Landmarks instead of Contours." 2020 IEEE International Conference on Bioinformatics and Biomedicine (BIBM). IEEE, 2020.

or 

Mozaffari, M. Hamed, and Won-Sook Lee. "Real-time Ultrasound-enhanced Multimodal Imaging of Tongue using 3D Printable Stabilizer System: A Deep Learning Approach." arXiv preprint arXiv:1911.09840 (2019).

Author Response

We thank referees for their comments and suggestions. The following revisions made accordingly:

As recommended, we expand the introduction giving more examples of deep learning application in various fields (lines 110 -130).

We also went through the text made minor corrections in typo, formatting as well as rewording some sentences for better expressing the ideas. These changes can be seen with MS-Word tracking feature.